# Seroprevalence of Cysticercosis among Epileptic Patients Attending Neurological Units in the Urban Area of Abidjan

**DOI:** 10.3390/microorganisms9081712

**Published:** 2021-08-11

**Authors:** Man-Koumba Soumahoro, Jihen Melki, Berthe Assi, Yves Landry Kangah, Mamadou Camara, Gildas Boris Tazemda-Kuitsouc, Mireille Nowakowski, Constance Yapo-Ehounoud, Thérèse Sonan, Jacques Bellalou, Ronan Jambou

**Affiliations:** 1Department of Epidemiology—Clinical Research, Institut Pasteur de Côte d’Ivoire, Abidjan 01 BP 490, Côte d’Ivoire; yveskangah@yahoo.fr (Y.L.K.); drkuitsouc@gmail.com (G.B.T.-K.); 2Department of Parasitology, Institut Pasteur de Côte d’Ivoire, Abidjan 01 BP 490, Côte d’Ivoire; jihenmelki@gmail.com; 3Neurology Department, Cocody University Hospital, Abidjan 01 BP V 13, Côte d’Ivoire; berthassi2000@gmail.com (B.A.); constanceyapoehounoud@gmail.com (C.Y.-E.); 4Neurology Unit, Adjamé General Hospital, Abidjan 03 BP 1856, Côte d’Ivoire; mamadouc8@gmail.com; 5Recombinant Protein Platform, Institut Pasteur, 75015 Paris, France; mireille.nowakowski@pasteur.fr (M.N.); jacques.bellalou@pasteur.fr (J.B.); 6Neurology Department, Yopougon University Hospital, Abidjan 21 BP 632, Côte d’Ivoire; sonantherese@gmail.com; 7Global Health Department, Institut Pasteur, 75015 Paris, France

**Keywords:** cysticercosis, urban area, epilepsy, seroprevalence, Cote d’Ivoire

## Abstract

Cysticercosis is one of the main causes of secondary epilepsy in sub-Saharan Africa. To estimate the seroprevalence of cysticercosis among epileptic patients, we conducted a cross-sectional study of patients attending neurology consultation in Abidjan, Côte d’Ivoire. **Methods**: Patients’ socio-demographic and lifestyle data were collected as well as blood samples for serological testing using ELISA and Western blot based on IgG antibodies detection. For qualitative variables comparison, Chi^2^ or Fisher tests were used; a Student’s t-test was used to compare quantitative variables. A multivariate logistic regression model was fit to identify risks factors. **Results**: Among 403 epileptic patients included in the study, 55.3% were male; the median age was 16.9 years; 77% lived in Abidjan; 26.5% were workers. Most patients included in the study had tonic-clonic seizures (80%), and 11.2% had focal deficit signs. The seroprevalence of cysticercosis was 6.0%. The risk was higher in patients over 30 years old (aOR = 5.1 (1.3–20.0)) than in patients under 16. The risk was also considerably high in patients who reported epileptics in the family (aOR = 5 (1.7–14.6)). The risk was three-fold less in females than in males. **Conclusions**: This study highlighted the exposure of epileptic patients to *Taenia solium* larvae in an urban area. The risk of positive serology was increased with age, male gender, and family history of epilepsy.

## 1. Introduction

Epilepsy is a common and chronic neurological condition that poses public health concern, especially in Africa [1]. According to the World Health Organization (WHO), it is one of the most widespread non-communicable diseases affecting about 50 million people worldwide [2]. The geographical distribution of this disease is uneven. The number of people suffering from epilepsy is much higher in low- and middle-income countries, where nearly 80% of cases are concentrated. Sub-Saharan Africa remains the most affected region, with an annual incidence ranging from 45.0 per 100,000 to 81.7 per 100,000. [2,3,4].

Studies conducted in the outpatient neurology departments of the Cocody University Hospital (Côte d’Ivoire) have shown that epileptic seizures are one of the main reasons for consultation [5].

Estimating the prevalence of each etiology of epilepsy can help design and develop a national program for the control of the disease. Parasitic infectious diseases such as cysticercosis, onchocerciasis and malaria are the main etiologies of epilepsy [6,7].

According to the World Health Organization, 30% of secondary epilepsy cases are due to cysticercosis in countries where the responsible parasite is endemic. Cysticercosis is caused by the tapeworm *Taenia solium* and is associated with fecal peril [8]. Cysticercosis is mostly prevalent in a rural environment where both free rearing of pigs and poor sanitation conditions are supposed to enhance transmission of the disease. However, very few studies have been conducted in urban areas. Indeed, in areas with informal houses, the sanitation conditions are poor, and the rearing of pigs is frequent. Additionally, many lone workers are used to eating street food where there is poor or limited hygiene. In this setting, a high level of transmission of cysticercosis can also occur. Cysticercus can develop in any organ, but most often, they reach the subcutaneous tissues, the eyes, and the brain, which can lead to vision dysfunction, seizures, and intracranial hypertension [9,10]. Neurocysticercosis (NCC) is the main cause of acquired epilepsy in tropical zones and is associated with 50,000 deaths yearly [10,11]. NCC often occurs where humans live closely together with pigs and are often considered predominant in rural areas. So far, there has been only one study by Schmidt et al. [12] that assessed the risk of NCC in urban environments, which found that although the prevalence is lower, NCC nonetheless occurs.

In Côte d’Ivoire, the data available on cysticercosis are out of date and inconsistent [6,13,14,15,16]. This first report addresses cysticercosis in an urban setting. In a second one, we will report data in rural conditions.

The objectives of this study are to describe the characteristics of epileptic patients diagnosed in neurology services of public hospitals in Abidjan, to estimate the seroprevalence of cysticercosis among these patients, and to analyze their exposure factors to cysticercosis.

## 2. Materials and Methods

### 2.1. Study Design and Setting

An exhaustive cross-sectional survey was conducted over an 18-month period (from September 2015 to February 2017) at public neurology services within the city of Abidjan, the economic capital of Côte d’Ivoire, in West Africa. It is the most populated city in the country, with a total area of 2119 km^2^ housing approximately 21% of the national population. The administrative division of this city is composed of two parts: Abidjan North and Abidjan South, on either side of the Ebrié Lagoon with ten communes. At the time of this study, the city of Abidjan offers a comprehensive health care services package at three public consultation services specialized in neurology located at the Adjamé General Hospital and within the University Hospitals Centres of Yopougon and Cocody. These three health structures are in Abidjan North, which is the most populated area of the city.

### 2.2. Recruitment of Patients

Patients were selected within the neurology units of the General Hospital Adjamé and the Cocody and Yopougon University Hospitals. The study population consisted of all individuals who visited a neurologist for the first time for epileptic seizures at one of the three sites cited above with patient or parental informed consent. The epileptic patients met the definition of the International League Against Epilepsy (ILAE): at least two spontaneous epileptic seizures within an interval of more than 24 h [17]. Patients whose etiology of epileptic seizures was already known (head injury, stroke, metabolic disorders, tumors) and those who refused to participate in the study were not included.

### 2.3. Data Collection

During regular consultation, neurologists approached patients who met the inclusion criteria to participate in the study. The patient was then referred to the investigator for an interview after giving their informed consent.

Data on patients’ socio-demographic characteristics, their environment, and their behavior concerning their hygienic habits were collected by a trained investigator during a face-to-face interview using a questionnaire. The investigators also collected the clinical characteristics as well as the type of treatments received by the patients. The radiological characteristics of the patients’ lesions were collected secondarily in the medical file with their consent.

### 2.4. Serology of Cysticercosis

Serological tests, ELISA and Western blot, were used to detect IgG antibodies against *Taenia solium* antigen in patient serum. ELISA and Western blot based on the glycosylated fraction of the cyst CS50 were conducted by the Department of Parasitology in the Institut Pasteur de Côte d’Ivoire, as previously described by Tsang et al. (1989) [18]. Antigens were obtained from cysticercus collected on infected pigs in slaughterhouses in Madagascar. The extraction of the antigens was conducted in Pasteur Institute in Paris, according to Tsang et al. [18]. Positive sera in the ELISA screening were further confirmed by Western blot. In the present study, positive serology for cysticercosis was defined as positive in both ELISA and Western blot tests.

Cysticercus of *T. solium*, collected from infected pig meat, were washed repeatedly in phosphate-buffered saline solution and stored at −80 °C until use. The procedure of extraction was described by Tsang et al. [18]. Briefly, 25 mL of cysticercus were disrupted in 75 mL extraction buffer then centrifugated at 5000× *g* for 30 min. The supernatant was centrifuged at 40,000 for 1 h and filtered through a Millipore filter with a pore size of 0.22 µm (Merck, Germany). The protein concentration was determined by the Bradford method. Antigens putification was carried out by affinity chromatography. After gel filtration on a Sephadex G-25M column, which was used to isolate proteins from crude extract, the latter was purified by affinity chromatography on a ConA-Sepharose 4B column (AKTA pure), equilibrated with a Tris-HCl buffer solution of pH 7.4 and composed of 0.5 M NaCl, 0.02 M Tris. It was then eluted with a 0.1 M α-D-methylglucoside solution. The protein peak was detected by spectrophotometry at 280 nm. The purified fractions of antigen were concentrated by ultrafiltration through a YM-10 membrane (Amicon Ultra).

For the ELISA test, we first coated 96-well microtiter plates (Nunc) with 100 µL per well of antigen CS50 diluted to a final concentration of 0.1 μg/mL in PBS (pH 7.4). The plates were incubated overnight at 4 °C then washed three times with PBS containing 0.05% Tween-20 (PBST) then blocked with 5% bovine serum albumin (150 µL/well) for 1 h at 37 °C. After washing the plates five times with PBST, we diluted a serum sample at 1∶200 in blocking buffer and added 100 µL to each well. After incubation for 2 h at 37 °C, the plates were washed three times with PBST. We added to each well a 100 µL of a diluted detection anti-Human IgG antibody solution conjugated to Horseradish Peroxidase. The plates were incubated for 1 h at 37 °C and washed again with PBST five times. Subsequently, 100 µL of o-phenylenediamine dihydrochloride (OPD) substrate solution was added to each well, and the plates were incubated for 20 min at 37 °C. Afterwards, the reaction was stopped by adding 50 µL of 2.5 N H2SO4, and the optical density (OD) was measured at 492 nm with the ELISA plate reader.

The enzyme-linked immunoelectrotransfer blot (EITB) assay was performed according to Tsang et al. (1989). To summarize, the proteins were transferred from the gel following SDS-PAGE onto a nitrocellulose membrane with a 0.2-μm-pore-size using Semi-Dry Electrophoretic system (Trans-Blot transfer system-Bio-Rad Laboratories, based in Hercules, California) for 1 h 30 at 30 V and 4 °C. The membrane was cut longitudinally, and each strip was blocked with 5% skim milk for 1 h. After washing three times with PBST, the strips were incubated with 1:20 diluted patient serum for 2 h and diluted 1:10,000 Alkaline Phosphatase conjugated anti-Human IgG for 1 h at 37 °C under constant shaking, respectively. After adding of 1 mL salt substrate of NBT/BCIP (nitro-blue tetrazolium chloride 5-bromo-4-chloro-3′-indolyphosphate p-toluidine), the plates were incubated for 30 min at 37 °C. We stopped the enzyme-substrate reaction by rinsing the strips with distilled water. Positive reactions were identified by the appearance of clearly defined bands.

### 2.5. Quality Control

The ring trial was organized by the European Cystinet consortium to evaluate all the methods used by public laboratories in the European Union. Series of anonymized serums were sent to all laboratories for a blind test. Institut Pasteur of Cote d’Ivoire was associated with this network and obtained 100% congruent results during each trial.

### 2.6. Data Analysis

A double entry of the collected data was carried out by two trained operators in the EpiData version 3.1 software (EpiData Association, Odense, Denmark) Data were then analyzed using the Stata^R^, version 11.0 software.

Quantitative variables (such as age) were described by their median (interquartile range: IQR) and qualitative variables by proportions. The primary outcome of the study was positive serology for cysticercosis in epileptic patients.

The search for the determinants of positive serology for cysticercosis in epileptic patients was initially carried out by univariate analysis. To do so, qualitative variables were compared in seropositive (CYSTI+) versus seronegative (CYSTI−) epileptic patients by the Chi^2^ or Fisher’s exact test, as appropriate, and quantitative variables were compared by the Student’s t-test or Wilcoxon test, as appropriate. Interaction or confounding factors were investigated by stratified analysis of age and gender factors. Logistic regression was then performed on the variables associated with the dependent variable (positive cysticercosis serology in epileptics) with a degree of significance *p* < 20%, using a stepwise descending procedure. The final model retained only the variables associated with the dependent variable at the 5% threshold in the bilateral formulation. Hosmer and Lemeshow’s goodness of fit test was used to verify the final model adequacy.

### 2.7. Ethical Considerations

All the participants or their parents were informed about the objectives and the procedures of the study. Written informed consent was obtained for each patient before inclusion. The study was approved by the National Ethics and Research Committee (CNER) of the Ministry of Health of Côte d’Ivoire on 29 May 2015 under the number 022/MSLS/CNER-dkn.

## 3. Results

### 3.1. Characteristics of Epileptic Patients

Among the 403 people included in the study, 55.3% were male and with a median age of 16.9 years (IQR: 9.3–28.3 years). More than half of the patients reside in the northern part of the city of Abidjan (57.6%), 47.2% of them were schoolchildren or students. Out-of-school accounted for the majority of the unemployed (59/102) or 15.2% of the total study population. Among adult patients, 34.5% reported living with a partner (Table 1).

The neurology department of the Cocody University Hospital included more cases (*n* = 258) than the one of Yopougon (*n* = 128) or the Adjamé General Hospital (*n* = 17).

According to the clinical examination performed by neurologists, 80% of the epileptic patients had tonic-clonic seizures, which were generalized in most cases (83.8%). Almost all these seizures had a motor manifestation (98.3%) and were triggered at any time (74.5%). A family history of epilepsy was revealed by 8.8% of the patients, while 3.8% of these patients reported a history of neonatal suffering and 5.3% reported head trauma. Neurologists noted focal neurologic deficits in 45 patients (11.2%).

Sodium valproate and phenobarbital were prescribed as monotherapy to 44.56% and 16.98% of the patients, respectively. Forty-one patients received dual therapy, and one was given triple therapy (phenobarbital, sodium valproate and clonazepam).

Electroencephalograms (EEG) were prescribed to 395 patients, but 346 (87.6%) of them reported having performed it. However, the EEG results were only available for 206 (52.2%) of these patients. Results were normal for 77 (37.4%) and abnormal for 129 (62.6%) of them. Abnormalities were generalized, synchronous, and symmetrical for 77/129 (59.7%) cases and lateralized for 48/129 (37.2%) cases. Three patients had other types of abnormalities: slow waves of irritative appearance or slow overloads.

CT scans were prescribed to 392 patients, and 227 (57.1%) reported to have done one. However, the results were available in their medical records for only 114 (50.2%). Results were normal for 76 (66.7%) epileptics patients. Of the 38 people with CT scans harboring abnormal images, 6 (15.8%) had lesions consistent with abscesses, 8 (21.1%) with vascular abnormalities, 2 (5.3%) with brain tumors, and 22 (57.9%) had non-specific lesions.

For the six patients with images related to abscesses, a CT scan was available after treatment. Three of them harbored calcifications of the lesions, and for the only one with evocating round lesions, the albendazole treatment-induced regression of the lesions (Figure 1). Thus, in line with Del Brutto et al. [19], in addition to the clinical signs (convulsions) and residence in an endemic area, the neuroimaging of these patients confirmed the diagnosis of neurocysticercosis.

Among the 403 patients included in the study, 34 (8.44%) had a positive ELISA serology for cysticercosis. This positive serology was confirmed with Western blot in 24 of them (70.9%). Thus, according to the case definition of cysticercosis in this study, 6% (95%CI: 4–8.7%) of the epileptic patients were seropositive for cysticercosis (Table 1).

### 3.2. Positive Serology Determinants in Epileptic Patients

Factors related to positive serology for cysticercosis in patients with epilepsy in univariate analyses were age (*p* = 0.007), marital status (*p* = 0.003) and presence of children in the home (*p* = 0.003) (Table 2).

Indeed, compared to the youngster, people aged 16 to 30 years and those over 30 years were more likely to have a positive serology for cysticercosis. Epileptic patients who reported living in a relationship were less likely to be positive (20.8% vs. 58.3%; *p* = 0.0003). Similarly, the presence of children in the home was linked to positive serology (58.3% vs. 41.8%; *p* = 0.003). Furthermore, univariate analyses showed a link between gender and seropositivity that had limited significance (*p* = 0.046), where men were more often seropositive than women (75% vs. 25%). Family history of epilepsy was also linked to a positive serology for cysticercosis (*p* = 0.008), suggesting family clusters as a determinant of seropositivity (Table 2). Among the epileptic patients with positive serology, 79.2% reported consuming pork, and the frequency of positive serology was high (*p* = 0.049). The cysticercosis positive serology of epileptic patients was independent of their living environment or reported hygienic habits (see Table 3).

However, the multivariate logistic regression model retained only associations between cysticercosis-positive serology and male sex and between age and a family history of epilepsy. Thus, women with epilepsy are three times less exposed to cysticercosis than men (ORa = 0.32; 95% CI (0.11–0.91)), and older epileptics were four to five times more likely to have been in contact with *Taenia solium* larvae than younger ones. Individuals who reported a family history of epilepsy were almost five times more likely to have a positive serology compared to those who did not (*p* = 0.003) (Table 4).

## 4. Discussion

The epileptic patients included in the study were mostly young people, as observed in several other studies conducted in sub-Saharan Africa [12,20,21,22]. Our data show a predominance of male patients, similarly to what has been previously reported in the literature [6,20,23,24]. Epilepsy is mainly induced at two periods during cysticercosis, i.e., at the beginning due to oedema and when the parasite dies and induces calcification.

Nearly half of the patients screened for the study were unable to perform CT scans due to limited financial resources. This limited the ability to specify the type and position of the lesion and to confirm their final diagnosis. These brain CT images included cystic and nodular-looking images and hypodensities, which can support several differential diagnoses, such as tuberculosis, brain abscess, brain tumors, and vasculitis [25]. For cysticercosis, the detection of the scolex as a shiny nodule within the cyst is a diagnosis of certainty and a criterion for cyst viability [26]. However, this typical aspect was not documented in this series, but other aspects are evocating.

According to Del Brutto’s revised NCC classification, diagnosis of NCC is based on a combination of neuroimaging, clinical examination, the notion of exposure, and immunological analysis [19]. Following these recommendations, during this study, only two cases of neurocysticercosis could be confirmed, which is quite similar to other studies. Studies of neurocysticercosis in an urban setting are scarce, such as that of Schmidt et al. in Dar es Salaam (Tanzania), who reported six cases of neurocysticercosis in a series of 306 registered epileptic patients, i.e., nearly 2% of them [12].

In the same line, negative Elisa serology was observed in some patients with calcified neurocysticercosis evocating CT scans images, which is quite confusing. Some authors have estimated the sensitivity and specificity of the ELISA serology test to be 62% and 70%, respectively [27,28]. However, at the calcified stage of the lesions, serology has a limited diagnostic value. More than 50% of NCC cases detected by imaging with calcified lesions have a negative serological result [29,30]. This underlines the importance of a combination of several criteria for the diagnosis of neurocysticercosis [19].

To the best of our knowledge, it is the first time that cysticercosis can be detected in epileptic patients in urban areas of Côte d’Ivoire. Data show a seroprevalence of 8.4% (34/403) with ELISA and a confirmation rate of 70.9% (24/34) with Western blot. According to our study’s case definition, 24 out of 403 (6.0%) epileptic patients were seropositive for cysticercosis. This frequency is lower than that reported in epileptic patients in Burundi (11.7% out of 103) and in Rwanda (21.8% out of 211) with Western blot and in Tanzania (15% out of 40) with the marketed LDBiodiagnostic kit [22,31,32]. The differences observed in seroprevalence may be due to the diagnosis methods used as well as the research study environments. In Latin American countries, Bruno et al. (2013) estimated median seroprevalence in urban areas to 12.1% among people with epilepsy based on Western blot-LLGP [33]. In an Indian study of epileptic patients living in a slum, Singh G and colleagues showed that the seroprevalence of cysticercosis was as high as 25.5% (27/106) [34].

With a three-fold higher risk for male compared to female patients, this study demonstrates an association between the seroprevalence of cysticercosis and gender in epileptic patients. The seroprevalence was greater in older patients, where the risk was four to five times higher than in younger patients. This observation was also demonstrated in a study conducted in Benin. Indeed, Houinato et al. (1998) determined that seroprevalence using Western blot was higher in men (1.9%) than in women (0.8%) and increased with age [35]. This difference can be due to calcification of the parasite but also to the lifestyle of older, which are more frequently alone and eat street food (inducing tape worm), or at least to a cumulative effect of infectious contacts. As already mentioned, this gender imbalance could be related to a higher transmission occurring in lone male urban workers. Some risk factors were also highlighted, such as family cases, which can be related to the family focus of cysticercosis transmission, especially if the tape worm carrier is inside the family.

Although our study was able to determine the seroprevalence of cysticercosis in epileptic patients in the Abidjan metropolitan area and to identify risk factors, it has some limitations. First, our questionnaire should have captured detailed information on the travel history to rural areas where poor sanitary conditions and the local pig breeding promote the circulation of *T*. *solium* in the population [36]. Second, Praet et al. suggest that infected pigs are probably excluded from the commercial chain in urban areas and have difficulty reaching urban markets and street kitchens, which explains the low exposure to cysticercosis [37]. However, this is not the case in Côte d’Ivoire, where street kitchens are mostly supplied directly from rural areas, probably due to the lower cost and local preferences.

Third, the Western blot-LLGP reference technique was used solely for seropositive epileptic patients with ELISA, suggesting that the seroprevalence of cysticercosis in epileptic patients could have been slightly underestimated.

## 5. Conclusions

This first study in the urban area in Côte d’Ivoire highlights cysticercosis transmission in this area. Two cases of neurocysticercosis were also diagnosed in this context.

The study estimated the seroprevalence of cysticercosis at 6% in epileptic patients consulting public neurology services in the city of Abidjan. The risk increased with age, male sex, and the notion of familial epilepsy. The frequency of positive cysticercosis serology in epileptics is moderate in the urban context in Côte d’Ivoire. Serological studies should be conducted in the suburbs, where pigs are also reared.

This study made the use of the cysticercosis serology test available for incoming patients at the Institut Pasteur of Cote d’Ivoire.

## Figures and Tables

**Figure 1 microorganisms-09-01712-f001:**
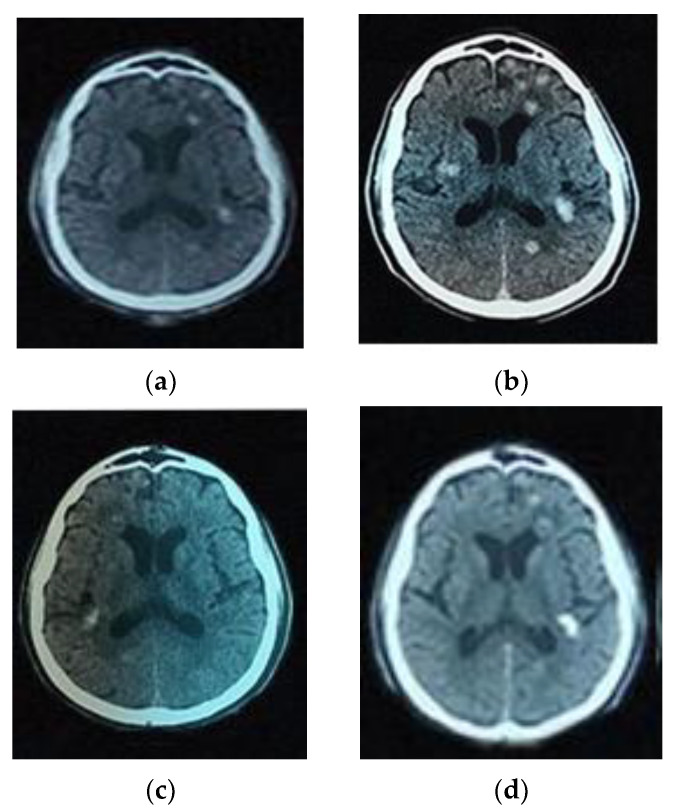
Brain CT scan images of a patient before and after treatment with Albendazole. (**a**) Image of the brain CT scan without contrast product, showing plurifocal nodular lesions reflecting cysticerci of different ages and some of which were calcified before treatment; (**b**) Multifocal nodular image more abundant after injection of the contrast product before treatment; (**c**) Resolution of some lesions with the persistence of calcifications after treatment with a cerebral scanner without contrast product after treatment; (**d**) Image of the brain scanner after injection of the contrast product after treatment.

**Table 1 microorganisms-09-01712-t001:** Characteristics of epileptic patients diagnosed in public neurology services of the city of Abidjan between September 2015 and February 2017.

Parameters	Items	N (%)
Gender (N = 403)	Male	223 (55.3)
	Female	180 (44.7)
Age group (N = 403)	<16 years old	164 (40.7)
	16–30 years old	150 (37.2)
	>30 years old	89 (22.1)
Living as a couple (N = 403)	No	127 (31.5)
	Yes	67 (16.6)
	NA	209 (51.9)
Presence of children in the household (N = 403)	No	70 (17.4)
	Yes	333 (82.6)
Housing (N = 383)	Abidjan North	213 (57.6)
	Abidjan South	78 (19.5)
	Outside Abidjan	92 (22.9)
Professional status (N = 388)	Workers	103 (26.5)
	Schoolchildren/students	183 (47.2)
	Unemployed	102 (26.3)
Living in urban areas (N = 379)	No	30 (7.9)
	Yes	349 (92.1)
Cysticercosis serology (N = 403)	Negative	379 (94.0)
	Positive	24 (6.0)

**Table 2 microorganisms-09-01712-t002:** Socio-demographic and clinical characteristics of epileptic patients included in the study by cysticercosis serological status.

Socio-Demographic Characteristics		CYSTI−	CYSTI+	*p*
Sex (N = 403)	Male	205 (54.1)	18 (75.0)	0.046 ^1^
	Female	174 (45.9)	6 (25.0)	
Age group (N = 403)	<16 years old	161 (42.5)	3 (12.5)	0.007 ^1^
	16–30 years old	138 (36.4)	12 (50.0)	
	>30 years old	80 (21.1)	9 (37.5)	
Leaving as a couple (N = 403)	No	113 (29.8)	14 (58.4)	0.003 ^1^
	Yes	62 (16.4)	5 (20.8)	
	NA	204 (53.8)	5 (20.8)	
Presence of children in the household (N = 403)	No	60 (15.8)	10 (41.7)	0.003 ^1^
	Yes	319 (84.2)	14 (58.3)	
Professional status (N = 388)	Workers	94 (25.8)	9 (37.5)	0.213
	School children/students	171 (47.0)	12 (50.0)	
	Unemployed	99 (27.2)	3 (12.5)	
Housing (N = 401)	Abidjan North	216 (57.3)		0.742
	Abidjan South	75 (20.0)		
	Outside Abidjan	86 (22.8)		
Muslim obedience	No	256 (69.4)	20 (83.3)	0.174
	Yes	112 (30.6)	4 (16.7)	
Family history of epilepsy (N = 397)	No	346 (92.3)	16 (72.7)	0.008
	Yes	29 (7.7)	6 (27.3)	
Fevered seizures in childhood (N = 399)	No	347 (92.3)	22 (95.6)	1.000
	Yes	29 (7.7)	1 (4.4)	
Results of Electroencephalogram (N = 206)	Normal	72 (37.1)	6 (50.0)	0.835
	General Symmetric Abnormalities	73 (37.6)	4 (33.3)	
	Lateralized Abnormalities/defects	46 (23.7)	2 (16.7)	
	Other	3 (1.6)	0 (0.0)	
Presence of focal deficit signs (N = 401)	No	334 (88.4)	22 (95.6)	0.494
	Yes	44 (11.6)	1 (4.4)	

^1^ Variables selected for the multivariate model (*p* < 20%).

**Table 3 microorganisms-09-01712-t003:** Hygienic habits and living environments of epileptic patients included in the study by cysticercosis serologic status.

Life Environment	Items	CYSTI−	CYSTI+	*p*
Access to drinking water (N = 402)	No	73 (19.3)	4 (16.7)	1.000
	Yes	305 (80.7)	20 (83.3)	
Sewer (N = 402)	Pipes (sink)	233 (61.6)	16 (66.7)	0.671
	Gutter/trough/street	145 (38.4)	8 (33.3)	
Environmental Hygiene (N = 401)	Poor	248 (65.8)	16 (66.7)	0.929
	Good	129 (34.2)	8 (33.3)	
Presence of pigs near the dwelling (N = 402)	No	361 (95.5)	23 (95.8)	1.000
	Yes	17 (4.5)	1 (4.2)	
Compliance with the three hand washing times (N = 397)	No	182 (48.8)	9 (37.5)	0.28
	Yes	191 (51.2)	15 (62.5)	
Type of water used for dishes (N = 402)	Pump water	325 (85.8)	20 (83.3)	0.719
	Well water	53 (14.02)	4 (16.7)	
Lunch place (N = 403)	House	262 (69.1)	16 (66.67)	0.800
	Outdoor	117 (30.9)	8 (33.33)	
Dinner place (N = 403)	House	364 (96.0)	22 (91.7)	0.268
	Outdoor	15 (4.)	2 (8.3)	
Pork consumption (N = 403)	No	156 (41.16)	5 (20.8)	0.049 ^1^
	Yes	223 (58.84)	19 (79.2)	
Use of toilet (N = 398)	No	17 (4.5)	0 (0.0)	0.61
	Yes	357 (95. 5)	24 (100.0)	

^1^ Variable selected for the multivariate model (*p* < 20%).

**Table 4 microorganisms-09-01712-t004:** Results of multivariate logistic regression analysis.

Independent Variables	Items	CYSTI– N (%)	CYSTI+ N (%)	ORa (IC à 95%)	*p*
Sex	Male	205 (54.1)	18 (75.0)	1	
	Female	174 (45.1)	6 (25.0)	0.32 (0.11–0.91)	0.033
Age groups	<16 years old	161 (42.5)	3 (12.5)	1	
	16–30 years old	138 (36.4)	12 (50.0)	4.31 (1.16–16.10)	0.030
	>30 years old	80 (21.1)	9 (37.5)	5.07 (1.28–20.01)	0.021
Family history of epilepsy	No	346 (92.3)	16 (72.7)	1	
	Yes	29 (7.7)	6 (27.3)	4.99 (1.70–14.61)	0.003

Hosmer-Lemeshow goodness of fit test: *p* = 0.4.

## Data Availability

The data that support the findings of this study are available from the authors upon request.

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
