# Peer review of "Seroprevalence of Cysticercosis among Epileptic Patients Attending Neurological Units in the Urban Area of Abidjan"

_microorganisms, 2021, doi:10.3390/microorganisms9081712_

Round 1
Reviewer 1 Report
Soumahoro et al. report on cysticercosis prevalence in epileptic patients as calculated based on serological assessments in urban Adidjan.
Although the topic is interesting, this reviewer has serious concerns about the methodology. The authors address the potential problem of lacking sensitivity of their laboratory-designed in-house serology assays. However, they do not address specificity of the applied assays with the exemption of a general specificity estimation of (any) T. solium ELISA in the range of 70% (discussion, line 306). However, due to considerable cross-reactivity between antibodies against various helminths, specificity is indeed an issue of relevance. Considering the low proportion of positive samples (i.e., a low pre-test-probability) in combination with unknown specificity of the applied in-house assays, cysticercosis prevalence might in fact have varied between zero and six per cent, making any interpretation of the results challenging.
Within the paper, no assessment of the test accuracy of the applied in-house serology assays is presented; not even an evidence-based cut-off value for the in-house ELISA. However, without such information, it is virtually impossible to interpret the results.
There are also some minor comments:
The conclusion of the abstract is more a summary of the study aims than a concluding remark.
Discussion, line 291: The fix association of brain abscesses and toxoplasmosis is difficult to understand.
Discussion, lines 294-295: Might the fact that typical CT-morphologies were missing indicate the fact that the brain lesions could have been due to other causes than Taenia solium as well?
Author Response
We would like to first thank the reviewer for his time and for the constructive comments he provided.
Point 1: Although the topic is interesting, this reviewer has serious concerns about the methodology. The authors address the potential problem of lacking sensitivity of their laboratory-designed in-house serology assays. However, they do not address specificity of the applied assays with the exemption of a general specificity estimation of (any) T. solium ELISA in the range of 70% (discussion, line 306). However, due to considerable cross-reactivity between antibodies against various helminths, specificity is indeed an issue of relevance. Considering the low proportion of positive samples (i.e., a low pre-test-probability) in combination with unknown specificity of the applied in-house assays, cysticercosis prevalence might in fact have varied between zero and six per cent, making any interpretation of the results challenging. Within the paper, no assessment of the test accuracy of the applied in-house serology assays is presented; not even an evidence-based cut-off value for the in-house ELISA. However, without such information, it is virtually impossible to interpret the results.
Response 1: A quality control test has been performed and we have included a paragraph explaining that in the method section: “Ring trial was organized by the European Cystinet consortium to evaluate all the methods used by public laboratories in European Union. Series of anonymized serums were sent to all laboratories for blind test. Institut Pasteur of Cote d’Ivoire was associated with this network and obtained 100% congruent results during each trial.” We also calculate the 95% confidence interval [4.0% - 8.7%] of the estimated seroprevalence of cysticercosis.
Point 2: The conclusion of the abstract is more a summary of the study aims than a concluding remark.
Response 2: We improve the conclusion of the abstract and strengthen the conclusion of the paper as follow: “This first study in urban area in Côte d’Ivoire highlights cysticercosis transmission in this area. Two cases of neurocysticercosis were also diagnosed in this context. The study estimated the seroprevalence of cysticercosis at 6% in epileptic patients consulting public neurology services in the city of Abidjan. The risk increased with age, male sex, and the notion of familial epilepsy. The frequency of positive cysticercosis serology in epileptics is moderate in urban context in Côte d'Ivoire. Serological studies should be conducted in suburbs, where pigs are also reared.
This study made the use of cysticercosis serology test available for incoming patient at the Institut Pasteur of Cote d’Ivoire.”
Point 3: Discussion, line 291: The fix association of brain abscesses and toxoplasmosis is difficult to understand.
Response 3: We did not mean to link the two. We just used the word “toxoplaxmosis” to provide an example of infection that could lead to brain abscess. In order to avoid confusing the reader, we removed it in the text.
Point 4: Discussion, lines 294-295: Might the fact that typical CT-morphologies were missing indicate the fact that the brain lesions could have been due to other causes than Taenia solium as well?
Response 4: Indeed, to rule out other causes, we referred to the Del Brutto classification which uses a combination of criteria to make the diagnosis of neurocysticercosis.
Reviewer 2 Report
Dear Authors
Concerning your manuscript microorganisms-1306165 “Seroprevalence of cysticercosis among epileptic patients attending neurological units in the urban area of Abidjan”, I believe it is an interesting and very important clinical issue regarding neurocysticercosis (NCC) everywhere, taken its global importance for humans as consumers of domestic and wild pigs, but also as an important food and waterborne disease. And this type of research/results should have more visibility, both on regional, but also on African and global levels, namely because it can have application for other areas and studies regarding Taenia solium/Cysticercus prevalence and their consequences in Public and Human Health, specifically regarding the neurological impact of this disease.
Besides what will be pointed out, namely that your manuscript has potential to be published, the final decision on the publication of your manuscript depends on the Editor final statement.
Regarding my reviews and comments, they are as follows:
Abstract
Page 1, Line 36 – Write urban area.
Key-words
Page 1, Line 37 – Write urban area.
- Introduction
Page 2, Line 68 – Write “…Schmidt et al. [12]”
- Materials and Methods
Page 3, Line 108 – Did you mean “…concerning their hygienic habits…” or “…concerning their hygiene…”?
Page 3, Line 125 – Write T. solium.
- Results
Page 5, Line 202 – Write “…than the one of Yopougon”.
- Discussion
Page 9, Line 336 – Write “…Praet et al. suggest…”.
- Conclusions
Page 9, Line 346 – Write “…urban area…”.
Page 9, Line 348 – Write “…where pigs are also reared”.
Final consideration – I appreciated a lot your English level, so I congratulate you for the proof reading of the manuscript.
Best regards and good luck with your amendments.
Reviewer
Author Response
We would like to first thank the reviewer for his time and for the constructive comments he provided.
Point 1: Page 1, Line 36– Write urban area. 

Response 1: The change was made in the text.
Point 2: Page 1, Line 37 – Write urban area.
Response 2: The change was made in the text.
Point 3: Page 2, Line 68 – Write “…Schmidt et al. [12]”
Response 3: The change was made in the text.
Point 4: Page 3, Line 108 – Did you mean “…concerning their hygienic habits…” or “…concerning their hygiene…”?
Response 4: The change was made in the text.
Point 5: Page 3, Line 125 – Write T. solium.
Response 5: The change was made in the text.
Point 6: Page 5, Line 202 –Write “…than the one of Yopougon”.
Response 6: The change was made in the text.
Point 7: Page 9, Line 336 – Write “…Praet et al. suggest…”.
Response 7: The change was made in the text.
Point 8: Page 9, Line 346 – Write “…urban area…”.
Response 8: The change was made in the text.
Point 9: Page 9, Line 348– Write “…where pigs are also reared”.
Response 9: The change was made in the text.
Reviewer 3 Report
see the attachment.

Author Response
We would like to first thank the reviewer for his time and for the constructive comments he provided.
Point 1: The authors should discuss possible NCC-related mechanisms leading to seizures and epilepsy. 

Response 1: Epilepsy is mainly induced at two period during cysticercosis i.e., at the beginning due to oedema, and when the parasite dye, and induce calcification.
Point 2: The authors found that “seroprevalence was greater in older patients where the risk was four to five times higher than younger patients”. The authors should discuss/speculate on the mechanisms underlying the higher seroprevalence in relatively older patients compared to younger patients. It is well known that younger brains are more susceptible to seizures than older brains.
Response 2: This difference can be due to calcification of the parasite, but also to the lifestyle older, which are more frequently lone and eaten street-food (inducing tape worm), or at least to a cumulative effect of infectious contacts.
Point 3: The risk was also higher when a family history of epilepsy is present. These findings need to be discussed.
Response 3: Some risk factors were also highlighted, like family cases which can be related to family focus of cysticercosis transmission especially if the tape worm carrier is inside the family.
Point 4: The seroprevalence of cysticercosis was 6%. Is this statistically significant?
Response 4: To check the statistical significance of the 6% seroprevalence in the study population, we calculated its 95% confidence interval: [4.0% - 8.7%]. Then the seroprevalence is statistically significant.
Point 5: The conclusions are weak and descriptive.
Response 5: We strengthen the conclusion as follow: “This first study in urban area in Côte d’Ivoire highlights cysticercosis transmission in this area. Two cases of neurocysticercosis were also diagnosed in this context. The study estimated the seroprevalence of cysticercosis at 6% in epileptic patients consulting public neurology services in the city of Abidjan. The risk increased with age, male sex, and the notion of familial epilepsy. The frequency of positive cysticercosis serology in epileptics is moderate in urban context in Côte d'Ivoire. Serological studies should be conducted in suburbs, where pigs are also reared.
This study made the use of cysticercosis serology test available for incoming patient at the Institut Pasteur of Cote d’Ivoire.”
Round 2
Reviewer 1 Report
The authors have adhered to my suggestions on a minimally acceptable level.
Reviewer 3 Report
The authors have been responsive to my comments and critiques.